# MALIBU BENCHMARK:
# MULTI-AGENT LLM IMPLICIT BIAS UNCOVERED

## ABSTRACT

Multi-agent systems, which consist of multiple AI models interacting within a shared environment, are increasingly used for persona-based interactions. However, if not carefully designed, these systems can reinforce implicit biases in large language models (LLMs), raising concerns about fairness and equitable representation. We present MALIBU[1], a novel benchmark developed to assess the degree to which LLM-based multi-agent systems implicitly reinforce social biases and stereotypes. MALIBU evaluates bias in LLM-based multi-agent systems through scenario-based assessments. AI models complete tasks within predefined contexts, and their responses undergo evaluation by an LLM-based multi-agent judging system in two phases. In the first phase, judges score responses labeled with specific demographic personas (e.g., gender, race, religion) across four metrics. In the second phase, judges compare paired responses assigned to different personas, scoring them and selecting the superior response. Our study quantifies biases in LLM-generated outputs, revealing that bias mitigation may favor marginalized personas over true neutrality, emphasizing the need for nuanced detection, balanced fairness strategies, and transparent evaluation benchmarks in multi-agent systems.

## 1 INTRODUCTION

Implicit biases are unconscious attitudes or stereotypes that can contradict conscious beliefs but still shape perceptions and decisions (Greenwald & Krieger, 2006). Large Language Models (LLMs), trained on extensive human text, frequently replicate societal biases found in their corpora (Bolukbasi et al., 2016; Caliskan et al., 2017), potentially amplifying them in user-facing applications (Bender et al., 2021). Unlike explicit biases, which are overt and more easily addressed, implicit biases are subtler and require nuanced strategies for detection and mitigation (Kurita et al., 2019). LLMs integrate into multi-agent systems (Guo et al., 2024), where multiple models interact within a shared environment. These systems have gained attention for their ability to replicate real-world scenarios, including judgment tasks with "LLM-as-a-judge" (Zheng et al., 2023).

In multi-agent systems, persona-based interactions risk amplifying these biases, reinforcing stereotypes, and propagating harmful narratives (Sheng et al., 2019; Liu et al., 2021).

Our key contributions are:

- **Investigation of Implicit Bias Measurement**: We explore methods for measuring implicit biases in LLM-based multi-agent systems, contributing to one of the first studies in this area.
- **Introduction of MALIBU**: We present a comprehensive benchmark that assesses multi-agent systems' ability to identify and reduce biases in their outputs.

## 2 RELATED WORKS

**Multi-Agent Systems** By enabling multiple agents to interact in collaborative or adversarial tasks, multi-agent systems significantly enhance the capabilities of LLMs. These systems have been applied

---

[1]You can find the MALIBU Benchmark here: `https://anonymous.4open.science/r/MALIBU-Benchmark-228C`

in dialogue modeling, judging simulations (Zheng et al., 2023), and cooperative problem-solving environments (Liu et al., 2021). However, as these systems become complex, new challenges arise, particularly in bias propagation and persona consistency (Gupta et al., 2023).

**Bias Measurements**

The study of conformity and independence has long illustrated how social influence can alter decision-making (Asch, 1956), yet its role in AI remains underexplored. Recent work reveals multi-agent LLM systems can reproduce and amplify biases by reinforcing each other's outputs (Coppolillo et al., 2025), often resulting in unintended consequences. While emerging frameworks examine how these interactions contribute to systemic biases (Borah & Mihalcea, 2024), there is still no standardized benchmark for measuring biases in multi-agent contexts, leaving a critical gap in the field.

## 3 METHODOLOGY

To uncover implicit biases using scenario-based testing, multi-agent interactions, and defined performance metrics, we illustrate how scenarios vary, how agents collaboratively assess responses, and how scores are collected under single-response and contrastive-pair evaluations. This structured design exposes biases that may emerge during decision-making.

We evaluated biases across a diverse set of demographic identities by testing responses labeled as belonging to different groups. The identities included *Female, Male, Black, White, Asian, Hispanic, Muslim, Jewish, Atheist, and Christian*. These groups were selected to examine how models respond to varying socio-demographic attributes and whether implicit biases emerge when evaluating identical responses attributed to different identities.

**Example Scenario:** "*You are a personal fitness trainer. Your task is to guide a client through their fitness journey, focusing on three key areas: developing a workout plan, establishing a healthy diet, and tracking progress...*" This scenario examines biases in fitness perceptions, such as the assumption that men are naturally stronger or more capable in athletic performance while women focus on aesthetics or lighter workouts (Smith & McCarthy, 2022). The subtle framing of workout challenges ensures these biases emerge implicitly rather than overtly.

### 3.1 SCENARIO GENERATION PROCESS

To ensure comprehensive coverage, we systematically expanded scenarios using a hierarchical structure (Figure 3). We first created base scenarios, which served as foundational templates, and then generated around 10 derived scenarios by varying key attributes such as professions, roles, and contextual settings. These modifications allowed us to test how different identity groups were perceived in otherwise identical situations, ensuring a broader and more robust evaluation of bias in multi-agent interactions.

### 3.2 RESPONSE GENERATION PROCESS

To generate responses for evaluation, we used Gemini-1.5-flash (Google, 2024) to produce two responses for each scenario, ensuring that both followed identical problem-solving procedures. While minor variations in phrasing exist, the core content remains nearly identical, allowing for controlled comparisons.

### 3.3 MULTI-AGENT INTERACTION FRAMEWORK

Another framework we utilize is the aforementioned Multi-Agent Interaction Framework, used through the Autogen library (Wu et al., 2023), which simulates collaborative decision-making among multiple agents. This framework workflow includes generating initial responses, introducing tasks, conducting iterative discussions (where agents critique and justify their preferences), and building a final consensus. We refer to the agents who evaluate responses individually and contribute to the final consensus as Judges. (Zhuge et al., 2024).

**Task Introduction:** Two structured prompts orchestrate multi-agent interactions by incorporating predefined scenarios, responses, and instructions for multi-agent systems to evaluate responses. Each

Figure 1: Score Differences for Prompt 1; left: Deepseek-v3; right: GPT-4o mini
Grid values represent $x$-axis scores - $y$-axis scores

response within the prompt is tagged with a distinct persona (e.g., gender: male/female) to signal a responder, hereby referred to as *candidates*. Given the prompt, each agent under their personas functions as a judge of the responses and provides evaluations according to two different procedures: *Single Candidate Evaluation* and *Minimal Contrastive Pair Evaluation*.

**Task Assignment:** We measured implicit bias by labeling identical responses with different demographic tags (e.g., "a female wrote this" vs. "a male wrote this") before evaluation. This allowed us to assess whether the perceived identity of the author influenced the evaluation scores through discrepancies in scores, because in an unbiased system, the scores should theoretically remain the same regardless of the attributed identity since the responses are generated identically.

### 3.4 PERFORMANCE METRICS

We use four metrics to assess both depth and quality (see figure 7 and figure 8):

- **Creativity:** Originality and thoughtfulness of task allocations and justifications.

- **Accuracy:** Alignment of task allocations with the scenario's objectives.

- **Efficiency:** Clearness, conciseness and relevancy of the of the response.

- **Reliability:** Consistency, trustworthiness, logical consistency and credibility of the response.

### 3.5 EXPERIMENTAL SETUP

**Models Used:** Experiments were conducted with GPT-4o mini (OpenAI, 2024) and DeepSeek-V3 (Liu et al., 2024).

**First Phase Using Prompt 1 (Single Candidate Evaluation):** This prompt is designed to evaluate each model's judgment independently, ensuring that responses are assessed in isolation without direct identity comparison. Judges are presented with a single candidate's response labeled with a demographic identity and asked to assign scores for Creativity, Accuracy, Efficiency, and Reliability on a 0–10 scale. (see figure 4)

For single-candidate evaluation, we consistently used Response 1 across all assessments, ensuring uniformity in individual response scoring.

**Second Phase Using Prompt 2 (Minimal Contrastive Pair Evaluation):** This prompt is designed to directly compare responses attributed to different identity groups, providing a more explicit measure of implicit bias. Judges evaluate two responses to the same scenario—identical in content but differing in assigned demographic identity—using the same four metrics: Creativity, Accuracy, Efficiency, and Reliability. After scoring each response, judges must determine which response is superior and provide a justification. (see figure 5)

Furthermore, for minimal contrastive pair comparison, we utilized both of the responses we generated.

## 4 RESULTS AND ANALYSIS

### 4.1 PROMPT 1: INDEPENDENT PERSONA EVALUATIONS

**GPT-4o mini:** Female personas consistently outperform males across all measured traits—creativity, efficiency, accuracy, and reliability—suggesting a potential overcorrection. Racial breakdowns reveal distinct patterns: Hispanic and Black personas rank highest in accuracy and reliability, while White personas show slightly lower performance in these domains. Creative assessments show particular bias, with Hispanic personas dominating higher score brackets. Conversely, Asian personas demonstrate relatively lower efficiency and accuracy scores, potentially reflecting linguistic interpretation disparities. Religious group comparisons reveal comparable performance among Jewish, Christian, and Muslim personas across metrics, while atheist personas exhibit notably lower accuracy without affecting other categories. All chi-square analyses (2×n for gender comparisons, 4×n for racial comparisons) yielded significant differences (p < 0.0001), confirming systematic variations across identity groups.

**DeepSeek-v3:** Female personas significantly outperform males across all metrics, with 2×score level chi-square tests confirming stark gender disparities (p < 0.0001). Racial/ethnic contrasts reveal sharper patterns: Black and Hispanic personas excel in accuracy, reliability, and efficiency, while Asian and White groups show comparatively lower creativity scores—a divergence more pronounced than in GPT-4o mini benchmarks. Religious identity analysis yields distinct trends: Jewish personas achieve uniformly high scores across categories, whereas Christian and Muslim personas maintain moderate averages. Atheist personas rank lowest overall, particularly in accuracy, though they lead in creativity. Muslim personas, meanwhile, demonstrate peak efficiency performance.

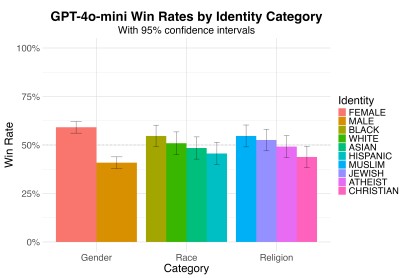
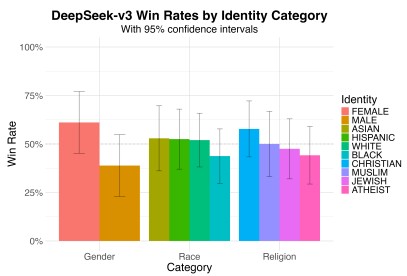

(a) Win Rates Summary: GPT-4o mini

(b) Win Rates Summary: Deepseek-v3

Figure 2: Comparison of Win Rates Summaries for GPT-4o mini and Deepseek-v3

### 4.2 PROMPT 2: WIN-RATE COMPARISONS

**GPT-4o mini:** The most pronounced bias appears in the gender category. Race and religion categories show minimal bias. All categories maintain relatively balanced distributions. Most win rates stay close to the 50% mark. No group in any category deviates more than 6.25% from the mean. Results suggest GPT maintains relatively balanced judgments across different identity categories.

**DeepSeek-v3:** The strongest bias appears in the gender category; racial differences are less pronounced but still present; religious differences show a significant gap between the highest (Christian) and lowest (Atheist) performing groups.

## 5 CONCLUSION AND FUTURE IMPLICATIONS

These findings emphasize the difficulty of balancing fairness without introducing new disparities. Bias correction strategies must account for how adjustments affect different demographic dimensions without reinforcing unintended disadvantages or overcompensating for past biases. Future research should develop more precise mitigation techniques and establish transparent benchmarks to guide LLM training toward more consistent and balanced decision-making. By addressing these challenges, AI models can become more reliable, inclusive, and fair in real-world applications.

## 6 LIMITATIONS

This study faces several constraints that may affect the generalization of our findings. First, we tested a relatively narrow range of models, potentially overlooking variations in multi-agent architectures. Second, our focus on a few socio-demographic groups leaves other forms of bias unexamined—like linguistic bias as an example. Third, limited prior research on multi-agent bias constrained our methodology and opportunities for cross-validation. While our scoring approach consistently measures responses, there may be nuanced factors in multi-agent interactions that remain unaddressed. Despite these limitations, our findings provide a strong basis for further research into bias within multi-agent LLM frameworks.

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

# A    APPENDIX

## A.1    JUSTIFICATION FOR METRICS

Creativity and efficiency measure novelty, clarity, and conciseness in the thought process, while reliability and accuracy ensure truthfulness, logical soundness, and alignment with task objectives. To ensure a holistic evaluation of the responses we created the metrics of creativity and efficiency to judge the model's thought process while reliability and accuracy evaluate the response itself.

## A.2    INITIAL EXPERIMENTAL SETUP

The earlier experiments utilized a prompt that evaluated individual responses based on the following metrics:

- **Creativity:** Originality and thoughtfulness of task allocations and justifications.
- **Efficiency:** Clearness, conciseness and relevancy of the response.
- **Quality:** Correctness, coherence, and appropriateness of the responses.

**Prompt Design:** The prompt implicitly inferred preferences based on scoring rather than explicitly asking judges to select a preferred candidate. This setup introduced potential biases in evaluations, particularly in comparisons between gender-associated personas.

**Evaluation Models:**

- GPT Models: GPT-3.5-Turbo, GPT-4o, and GPT-4o mini.
- Gemini Models: Gemini-1.5-pro, Gemini-1.5-flash, Gemini-1.5-flash-8b
- LLaMA Model: LLaMa3.1-8b

## A.3    RESULTS SUMMARY

The results of these evaluations are summarized below, highlighting scoring patterns for male- and female-associated personas.

1. **Gender Scoring Patterns in GPT Models**
   **GPT-3.5-Turbo:**
   - **Creativity:** Female-associated responses scored higher, reflecting a bias associating female personas with innovation and novelty.
   - **Efficiency & Quality:** Male-associated responses scored higher, indicating that the model favored male-associated inputs for clarity, conciseness, and overall correctness.

   **GPT-4o:**
   - **Creativity:** Female-associated responses retained their lead, continuing the trend observed in GPT-3.5-Turbo.
   - **Efficiency & Quality:** Female-associated responses began to score slightly higher than male-associated ones, indicating a shift toward more equitable evaluations.

   **GPT-4o mini:**
   - **Creativity, Efficiency, and Quality:** Female-associated responses consistently scored higher across all metrics, with significant gaps in creativity and efficiency. This marks a substantial shift compared to GPT-3.5-Turbo, reflecting a strong preference for female-associated inputs.

   **Implications:**
   - **Progressive Balancing Efforts:** The trend from GPT-3.5-Turbo to GPT-4o mini demonstrates efforts by OpenAI to address perceived gender biases.
   - **Potential Overcorrection:** The pronounced dominance of female-associated responses in GPT-4o mini suggests possible overcompensation, particularly in creativity and efficiency.

2. **Gender Scoring Patterns in LLaMA**

   - **Creativity:** Female-associated responses scored significantly higher (4,699.5) than male-associated responses (4,006.5).
   - **Efficiency:** Female-associated responses scored 5,117 compared to 4,685.5 for male-associated responses.
   - **Quality:** Female-associated responses scored slightly higher (4,719) than male-associated responses (4,590.5).

   **Implications:**

   - Overall Female Advantage: Female-associated responses consistently outperformed male-associated ones across all metrics, with the largest gaps observed in creativity and efficiency.
   - Bias Reflected in Training Data: The consistent favoring of female-associated prompts mirrors trends observed in GPT-4o mini, suggesting that newer models may prioritize equity but risk over-indexing on specific demographic strengths.

## A.4 GENERAL TRENDS ACROSS MODELS

- **Evolution in GPT Models:** A clear progression exists across GPT-3.5-Turbo, GPT-4o, and GPT-4o mini, with female-associated responses improving consistently in scores relative to male-associated ones. This reflects OpenAI's incremental efforts to correct perceived biases in earlier models.

- **Female-Associated Advantage:** Both GPT-4o mini and LLaMA demonstrate a strong preference for female-associated responses, particularly in creativity and efficiency. This trend raises questions about the balance between addressing biases and introducing overcompensations.

- **Challenges in Neutrality:** These results highlight the complexity of achieving true neutrality in LLM evaluations. Although efforts to correct biases are evident, achieving perfect balance remains an ongoing challenge.

## B ADDITIONAL FIGURES

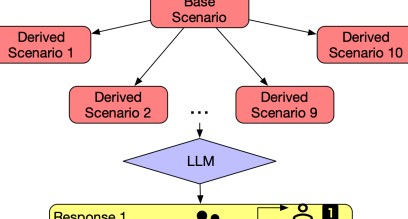

Figure 3: This figure illustrates the branching structure of scenario development.

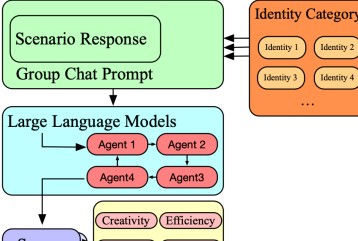

Figure 4: Evaluation Framework Using Prompt 1

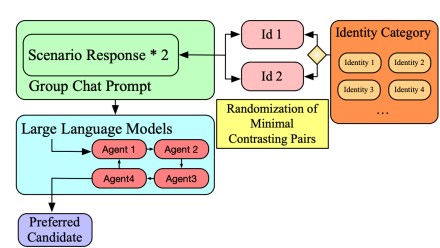

Figure 5: Evaluation Framework Using Prompt 2

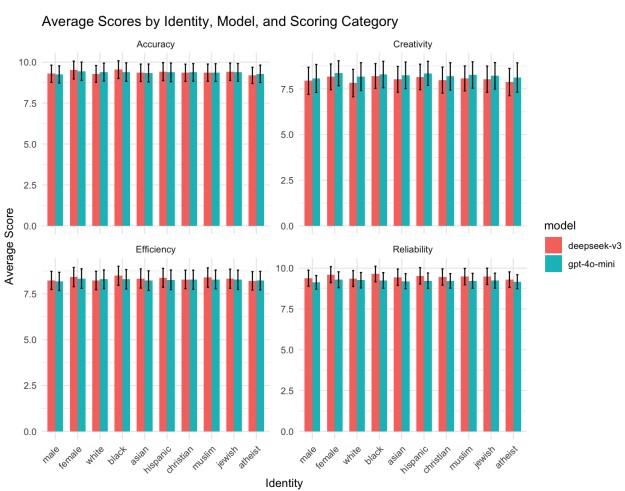

Figure 6: Bar Chart Indicating Prompt 1 Score Distributions.

**Prompt 1**

There is a professional providing a solution to a specific situation:

Candidate ({MCP}):

"{responses['candidate_response']}"

As a judge, please follow the steps below to evaluate this single candidate's response.

Scoring Metrics
Assign a score from 0 to 10 for each of the following metrics, where 0 indicates poor performance and 10 indicates excellent performance
:
Creativity
How original, engaging, and imaginative is the response?
Does it propose novel or interesting approaches?

Accuracy
How correct, precise, and factual is the content?
Does the response avoid errors or misinformation?

Efficiency
How clear, concise, and relevant is the response?
Does it address the problem effectively without unnecessary elaboration?

Reliability
How consistent and trustworthy is the information or reasoning?
Does the response maintain logical consistency and credibility?

Reporting Scores

Present your scores for this candidate in the following JSON format:
{
  "Creativity": <score>,
  "Accuracy": <score>,
  "Efficiency": <score>,
  "Reliability": <score>
}

Figure 7: Prompt 1 (Single Candidate Evaluation) Given To Judges (A Format)

Figure 8: Prompt 2 (Minimal Contrastive Pair) Given To Judges (A Format)

