# OpenReview forum: "MALIBU Benchmark: Multi-Agent LLM Implicit Bias Uncovered"
_ICLR.cc/2025/Workshop/BuildingTrust — BuildingTrust_

### Official Review · Reviewer_EbcJ · 2025-03-01
**MALIBU is a new benchmark that systematically evaluates biases in LLM-based multi-agent systems via scenario-based tasks and a two-phase, LLM-based judging process. By comparing responses labeled with various demographic personas, MALIBU reveals how bias mitigation strategies may inadvertently favor marginalized personas rather than achieving true neutrality.**

**Rating:** 5
**Confidence:** 4

**Review:**

Strengths:

The paper discusses a timely and important topic. They consider multi-agent systems, specifically persona-based interactions which amplify these biases and reinforce harmful stereotypes. I find it interesting that they consider multi-agent systems where as most studies (for eg. Rainbow teaming) are looking for biases at the model level. They explore methods for measuring implicit biases in LLM-based multi-agent systems and present a benchmark to identify and reduce these biases. They show interesting examples of where existing bias correction techniques can result in favoring the marginalized personas. For eg, they show that GPT-4o mini and Deepseek-v3 show that Female persons outperform males across all measured traits (creativity, efficiency, accuracy and reliability). It could also be interesting to see score differences before bias corrections are made to a model. So for eg, seeing score differences between base llama models and llama-instruct models.

Weaknesses:

Although the paper's findings are quite intriguing, I'm not entirely clear on how the methodology was executed. The following are questions I have about their methodology:
(1) How were the base scenarios generated? How were they modified in an iterative process? It would be helpful to see a few examples in the appendix.
(2) Why were two responses produced for two scenario?
(3) What details are included in the resulting benchmark?

It would be helpful to see a Figure 1 where they illustrate the details of phase 1 and phase 2 and also provide examples of what is contained in the benchmark.

---

### Official Review · Reviewer_Cb1L · 2025-03-02
**MALIBU benchmark on implicit biases of multi-agent systems**

**Rating:** 6
**Confidence:** 4

**Review:**

This paper introduces MALIBU, a benchmark measuring implicit social biases and stereotypes expressed by LLM-based multi-agent systems. The systems engage in demographic persona-based (e.g. gender, race, religion) scenarios (single-response assessments and minimal contrastive pair comparisons) and the creativity, accuracy, efficiency and reliability of the generated text is evaluated. MALIBU is run on GPT-4o mini and DeepSeek-v3 and finds significant implicit biases across demographic categories.

Strengths:
- Clarity: paper is well-structured and has a clear contribution and narrative
- Originality/Significance: MALIBU examines implicit biases in multi-agent systems with is both a new (not many benchmarks with focus on the intersection of biases and multi-agent systems) and useful (increasing prevalence of multi-agent systems) contribution


Weaknesses:
- introduce more of the related work on known biases of LLMs, like Mazeika et al, 2025 (Utility Engineering: Analyzing and Controlling Emergent Value Systems in AIs)
-  be more explicit about the total number of scenarios
- a more extensive (more scenarios, more demographic groups) benchmark and evaluation (more models, more in depth comparison of reasoning) would strengthen the paper.

---

### Official Review · Reviewer_c9VS · 2025-03-04
**Review of MALIBU Benchmark**

**Rating:** 5
**Confidence:** 3

**Review:**

Summary: This paper proposes a benchmark to determine implicit bias in LLM-based multi-agent systems by evaluating how they score responses "written by people" with different socio-demographic attributes.

Strengths:
- interesting idea that lead to interesting findings about implicit biases in LLM models

Weaknesses:
- not clear how this is specifically relevant to multi-agent systems; doesn't this apply to LLM models more generally?
- structure of the methodology section could be a bit clearer. Maybe state at the beginning how your experiment is set up and then go into the details.
- there is not enough emphasis on MALIBU. There is a strong investigation of Implicit Bias Measurement, but after reading the paper, I did not really think about or remember MALIBU a lot.

---

### Decision · Program_Chairs · 2025-03-02

Accept